

# The Copernicus Surface Velocity Platform drifter with Barometer and Reference Sensor for Temperature (SVP-BRST): Genesis, design, and initial results

Paul Poli[1], Marc Lucas[2], Anne O'Carroll[3], Marc Le Menn[4], Arnaud David[5], Gary K. Corlett[6], Pierre
Blouch[7], David Meldrum[8], Christopher J. Merchant[9], Mathieu Belbeoch[10], Kai Herklotz[11]

[1]Météo-France Centre de Météorologie Marine, Brest, 29200, France
[2]Collecte Localisation Satellites, Ramonville Saint-Agne, 31520, France
[3]European Organisation for the Exploitation of Meteorological Satellites, Darmstadt, 64295, Germany
[4]Service Hydrographique et Océanographique de la Marine, Brest, 29200, France
[5]NKE Instrumentation, Hennebont, 56700, France
[6]University of Leicester, Leicester, LE1 7RH, United Kingdom
[7]Retired from Météo-France, Plouzané, 29280, France
[8]Scottish Association for Marine Science, Oban, PA37 1QA, United Kingdom
[9]University of Reading and National Centre for Earth Observation, Reading, RG6 6AH, United Kingdom
[10]WMO-IOC Joint Technical Commission for Oceanography and Marine Meteorology in-situ Observing Programmes Support Centre, Plouzané, 29280, France
[11]Bundesamt für Seeschifffahrt und Hydrographie, Hamburg, 20359, Germany

*Correspondence to*: Paul Poli (paul.poli@shom.fr)

**Abstract.** To support calibration and validation of satellite Sea-Surface Temperature (SST) retrievals, over 60 High Resolution SST (HRSST) drifting buoys were deployed at sea between 2012 and 2017. Their data record is reviewed here. It is confirmed that sea-state and immersion depth play an important role in understanding the data collected by such buoys and that the SST sensors need adequate insulation. In addition, calibration verification of three recovered drifters suggests that the sensor drift is low, albeit negative at around -0.01 K/year. However, the statistical significance of these results is limited, and the calibration procedure could not be exactly reproduced, introducing additional uncertainties into this drift assessment. Based on lessons learnt from these initial buoys, a new-generation drifter was designed to serve calibration of SST retrievals by European Union's Copernicus satellites. The novel drifter includes an HRSST sensor calibrated by a metrology laboratory. The sensor includes a pressure probe to monitor immersion depth in calm water, and acquires SST data at 1 Hz over a 5-minute interval every hour. This enables the derivation of mean SST as well as several percentiles of the SST distribution. The HRSST sensor is calibrated with an uncertainty better than 0.01 K. Analysis of the data collected by two prototypes deployed in the Mediterranean Sea shows that the buoys are able to capture small-scale SST variations. These variations are found to be smaller when the sea-state is well-mixed, and when the buoys are located within eddy cores. This affects the drifter SST data representativeness, which is an aspect of importance for optimal use of these data.



## 1 Introduction

The Earth Observation Copernicus Sentinel programme, funded by the European Union, Iceland, and Norway, has driven the development of new space-borne sensors, with new ground segments and data processing chains. Of particular interest to oceanographers is the acquisition of high quality sea surface temperature (SST) data. Over short time scales, this essential

ocean state variable provides important information on the spatial distribution and intensity of dynamic structures, such as eddies, coastal currents and upwelling regions, in near real time (within a few hours after acquisition). Over the long term (multi-decade), it describes the distribution of heat within the Earth system. Long time-series of SST datasets (e.g., Merchant *et al*, 2014) are crucial to provide information on global and regional sea surface temperature trends. These can be used directly to monitor the evolution of the surface ocean on decadal time scales and help quantify the intensity of events such as

El Niño/La Niña, as well as being useful to constrain climate reanalyses (e.g., Dee *et al*, 2014). For these reasons, the importance of monitoring SST was recognized as a priority by the Copernicus programme, and a sensor aimed at observing SST was included on Sentinel-3 satellites, the Sea Land Surface Temperature Radiometer (SLSTR, Coppo *et al*, 2013). To deliver the SST data product service (Bonekamp *et al*, 2016), the dual-view capability and onboard calibration of SLSTR gives it greater accuracy than earlier generations of similar sensors, such as the Advanced Along-Track Scanning

Radiometer (AATSR, Llewellyn-Jones *et al*, 2001).

Satellite sensors measure top-of-atmosphere radiance, which has some relation to but is not identical to the physical temperature of Earth's emitting surface. The inverse process of inference of the surface state tends to amplify uncertainty. Achieving the desired quality of Earth Observations measurements from SLSTR places stringent requirements on the

SLSTR sensor calibration (Donlon, 2011), at a higher level than earlier generations of sensors. This drives a requirement for higher accuracy and better knowledge of uncertainties of the surface measurements used for validating the satellite products. This process requires the highest-possible quality *in-situ* measurements, with well-characterized uncertainties, so that the error budget of SST products can be investigated (e.g., Corlett *et al*, 2014). Such investigation requires covering the various regimes of satellite SST retrievals, mandating in turn that the high-quality *in-situ* data be geographically well-distributed.

As a result, concomitantly to the SLSTR development, the Copernicus programme aims to develop Fiducial Reference Measurement (FRM) initiatives. Among them is the deployment of an array of temperature measuring surface drifters, covering several SST regimes. The operational nature and climate quality of Sentinel-3 datasets are expected to deliver long-term data-records (Donlon, 2011). For consistency, this implies that the surface references used for calibration and validation

must also be homogeneous over time. This FRM initiative complements others started lately, such as under the European Space Agency (ESA) project Fiducial Reference Measurements for validation of Surface Temperature from Satellites (FRM4STS), which has conducted in particular a comparison of infrared radiometers with radiation thermometers in laboratory (Theocharous *et al*, 2018). Beyond comparisons, the goal is to establish the traceability of the various sensing





techniques to the Systeme International (SI) unit, as it then guarantees anchoring to international physical standards. In such attempt, the importance of metadata to define exactly the sensor and its environment is essential. For drifters measuring SST, this means knowing in particular the SST sensor depth and type, its calibration process, and other aspects influencing the buoy behaviour (such as drogue loss).

Based on lessons learnt from previous similar initiatives, a new type of drifter has had to be developed and submitted to a rigorous calibration procedure to meet this goal. In short, this new type of drifter must carry a state-of-the-art digital temperature sensor coupled to a hydrostatic water pressure sensor, allowing for a measurement frequency of up to 1 Hz. The value of this new drifter for calibration and validation (cal/val) of SST satellite retrievals is expected to be assessed through

international collaboration.

The outline of this paper is the following. Section 2 revisits the past HRSST drifting buoy initiatives, including error budget analysis. Based on the lessons learnt, Section 3 presents the design adopted for a new generation of drifter, called the Surface Velocity Platform drifter with Barometer and Reference Sensor for Temperature (SVP-BRST). Section 4 shows preliminary

measurement results from two SVP-BRST prototypes deployed in the Mediterranean Sea. Finally, Section 5 gives conclusions and prospects for future work.

## 2 Genesis: lessons learnt from past HRSST drifting buoy initiatives

### 2.1 Background: The HRSST-1 and -2 requirements

O'Carroll *et al* (2008) compared SST retrievals from AATSR with SST retrievals from a microwave sensor and with *in-situ*

SST from drifters. The drifters were found to have a standard deviation of error smaller than the microwave SSTs and larger than those from the AATSR. This highlighted the need for improved *in -situ* calibrated reference temperature data for satellite SST cal/val, particularly in reference to the validation of high-quality dual-view satellite SSTs, and the satellite and *in-situ* communities started a dialogue on collaboration and improvements. In 2009, the Group for High-Resolution SST (GHRSST) called on the Data Buoy Cooperation Panel (DBCP) HRSST Pilot Project (HRSST-PP) to implement a number

of key requirements for buoys to be eligible to support HRSST work (Donlon, 2009). The buoys would have to provide: hourly measurements, nominal or design depth in calm water of the drifting buoy SST to an absolute accuracy of 5 cm, location accuracy of 500 m, SST with a nominal resolution of 0.01 K or less and a total uncertainty of 0.05 K, and measurement time to within 5 minutes.

These requirements were adopted on a number buoys deployed by the Economic Interest Group (EIG) EUMETNET Operational Service for surface marine observations (E-SURFMAR) and European partners. This brought about four major technical improvements, as compared to standard practices at the time.



First, the location accuracy was increased, thanks to GPS instead of Argos for estimating position, and several buoys adopted Iridium instead of Argos for the transmission, to ensure regular hourly data reports. Second, the temperature was reported and transmitted to shore at a resolution of 0.01 K. These technical improvements are collectively known as 'HRSST-1'.

While only few buoys adhered to the HRSST-1 requirement in 2009, it has now become the standard, at the time of writing, for almost all drifters deployed globally. From there, a third requirement appeared, namely the adoption of a new Binary Universal Form for the Representation of meteorological data (BUFR) template in 2015, to encode the SST data at the resolution of 0.01 K, and transmit to operational data users via the World Meteorological Organization (WMO) Global Telecommunications System (GTS), without loss of information. That template became operational at most data originating

centers by the end of 2016: before that, many data transmitted on the GTS were sent at reduced SST resolution of 0.1 K. At the time of writing, all these three improvements are standard for most operational drifters.

The fourth technical improvement was for each buoy to use an individually-calibrated temperature probe, instead of one picked from a batch calibration, in order to guarantee the more stringent total uncertainty requirement of 0.05 K, as well as

traceability to national standards. This requirement (on top of previous ones) was called 'HRSST-2'. In total, 46 such HRSST-2 buoys fitted with all three technical advances, as well as including each a barometer, were deployed between 2012 and 2017. These buoys are listed in Table 1 below. They were manufactured by Metocean (Petolas, 2016), using Yellow Springs Instrument Company (YSI Inc.) sensors described in the table. One buoy was redeployed after running ashore.

In addition, several other HRSST-2 buoys were manufactured for experimental purposes, also by Metocean. Each buoy carried a Conductivity-Temperature (CT) probe manufactured by Sea-Bird Electronics (SBE) in order to measure salinity. Each HRSST-2 SVP buoy with Barometer and Salinity (SVP-BS) hence included two individual-calibrated SST probes: one integrated with the buoy hull (around 17 cm depth), and one in the CT probe (around 45 cm depth). This twin-sensor configuration offered near-optimal horizontal and temporal co-location by virtue of the buoy design. The only major

differences between the two sensors were the vertical positioning and the housing of the sensors (one digital SST sensor integral with the hull, the other CT sensor immersed entirely in water). In total, there were 19 such buoys deployed between 2012 and 2015 (one buoy was redeployed after beaching). Table 2 shows the list of such buoys, the deployment areas, and the mission dates. Most buoys were deployed in the North Atlantic.

## 2.2 HRSST-2 SVP-BS data record revisited

In order to exploit the co-located information from two individually-calibrated SST probes, the data record from the second set of HRSST-2 buoys, SVP-BS fitted with CT probes, is addressed here. The record consists of about 87,000 data reports between 2012 and 2016. Figure 1 shows a scatter density plot of the two temperatures. The twin measurements are highly correlated, and the robust standard deviation of the difference is 0.03 K. This result is compatible with uncertainty in a





difference of two sensors with total uncertainties better than 0.05 K (or possibly 0.02 K). However, Figure 1 shows a small fraction of outliers in both directions, especially for warmer temperatures. In fact, the Root Mean Square (RMS) of the differences is quite large, at 0.36 K.

The differences between the two measurements are not only due to sensor accuracy but also to the placement of the sensors: vertical location and housing (one integral with the buoy hull, the other underneath the buoy). To better understand the sources of differences, Figure 2a shows the differences between the two sensor temperatures as a function of solar elevation angle. As expected, most large-magnitude differences are positive during day time (the hull sensor being located closer to the surface). There are fewer large-magnitude differences are smaller at night and these are smaller when the Sun is more than

30 degrees below the horizon. The large departures are observed sometimes during day-time suggest that one or other of the two SST sensors may have been differentially affected by direct solar radiation, or by the buoy heating up the sensor through heat conduction.

Unlike promising new developments with wave drifters (Centurioni *et al*, 2016), the HRSST-2 drifters did not provide any

information about sea-state. Such information can be obtained however by co-locating with the ERA5 reanalysis (Hersbach and Dee, 2016). The ERA5 reanalysis data are interpolated in space from their original resolution (spectral truncation T639) to the buoy locations, using the nearest-in-time hourly reanalysis map. Figure 2b shows that the large-magnitude SST difference mostly arise when the sea-state is calm (significant wave heights under 2—3 m). The agreement between the sensors increases when there is more wave activity, probably because of greater mixing. When such is the case, almost all

SST differences are found in the range from -0.1 to 0.0 K. Sea-state mixing caused by waves cannot be controlled or mitigated by a platform as small as a 40 cm diameter drifter. However, the role of the waves, probably via mixing, is suggested here to be quite important when using the SST data collected by drifting buoys. A knowledge of the local SST dynamics, as the buoy is following a pendulum movement and senses the temperature surface at various depths within the top few metres of the ocean, would help better understand the distribution of SST that is measured, and how it corresponds

to satellite measurements, or how it should be considered in the cal/val process.

The differences between the probes can also be inspected as a function of mean solar local time (MSLT), for each buoy. For this, we only retain the buoys that reported at least for 250 days, without issue. For the subsequent data analysis, we filter out 12 cases when differences are larger than 20 K (visible in Fig. 1), likely to be erroneous. Figure 3 shows that the mean

differences feature a diurnal cycle, with the maximum positive differences around 12:00 MSLT. This is consistent with the depth difference of the two probes in the context of diurnal vertical stratification of the surface temperature. Diurnal stratification tends to peak around 14h (e.g., Morak-Bozzo *et al*, 2016), and temperature stratification larger than 0.1 K within the upper 0.5 m would tend to occur only at the lowest wind speeds. However, this daily cycle in difference may also be partially explained by the hull sensor being heated by the surrounding buoy, and/or by direct solar radiation (an effect

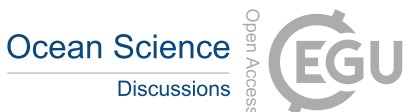

which might tend to peak more around 12h MSLT). These latter effects are not related to the environment and should be avoided.

## 2.3 Recovered buoys

Three HRSST-2 buoys manufactured in 2012, deployed in 2014, ran ashore in 2016 in Great Britain and Brittany. They were
recovered and offered together a unique opportunity to re-assess sensor accuracy and drift several years after initial calibration. The buoys were recovered without visible outer damage. It is not impossible that the sensors may have aged differently during the various phases of the buoy life cycle: (a) after calibration and until deployment, (b) at sea, (c) after recovery. Unfortunately, it proved impossible to have the probes calibrated by the same laboratory (Bernie Petolas, 2016, private communication). Despite the same Metocean interface being used at all three laboratories, the calibration procedure,
being inherently laboratory-dependent, brings in additional uncertainties. For example, the various laboratories involved here did not use the same verification points. The initial laboratory used three calibration points (0°C, 25°C, 40°C), i.e., the bare minimum to compute the three Steinhart-Hart coefficients per sensor. The same temperatures were then used to assess the (residual) calibration error. In the table, Lab.#2 refers to the Service Hydrographique et Océanographique de la Marine (SHOM) metrology lab, which used seven verifications points (between 2°C and 32°C, at steps of 5°C), and Lab. #3 refers to
the Scottish Association for Marine Science (SAMS) metrology lab, which used three verification points (0°C, 10°C and 20°C).

Table 3 shows the results of the calibration verification done by the initial laboratory (Measurement Specialties, Lab. #1 in the table), and the calibration verifications done by two other laboratories (at different dates), after the buoys were recovered
from shore. To remove the impact of different dates, the last column shows the estimated temporal drifts. The drift results vary in magnitude between the probes and the laboratories. This is probably mainly because of different choices for the verification temperatures, though other factors may have also played a role, such as probe resolution, probe response time, and temperature laboratory influence on the measurements (with the electronics not immersed in water), among others. However, all the results found here suggest negative trends, around -0.01 K/year for Lab. #2 and -0.005 K/year for Lab. #3.

Note, it cannot be ruled out that the probes, once removed from the buoys, did respond differently than during the initial calibration setup. Indeed, the temperature variations being looked at are very small, and any influence of the acquisition electronics may affect the results. The exact environment used for housing the electronics during calibration of the initial probes, as well as during the verifications, even if specified in the initial calibration sheets, cannot be replicated with
certainty.

Consequently, these results are to be taken with caution, and the importance of the calibration apparatus stands out as being an important part of the traceability. However, should the negative trend (cooling) be confirmed, it would have an impact on





the exploitation of the SST drifter data for satellite cal/val, as well as corrections that are made to global datasets. Recent adjustments have actually recognized buoys as being cooler than ships in terms of SST (Huang *et al*, 2015), though no difference was made especially for drifting buoys as a function of their 'age'.

In conclusion, given the importance of drifting buoy SST in climate studies, the impossibility of putting together firm metrology results indicates that a better-documented calibration protocol is needed for the measurement of SST by these platforms, both to ensure initial calibration and calibration verification several years afterwards.

## 2.4 Evaluation of HRSST-1 and HRSST-2 drifters

The analysis of O'Carroll *et al* (2008) identified the standard deviation of error of the drifting buoy network to be 0.23 K. An
interpretation of this finding is it is equivalent to the standard uncertainty of the error distribution. An alternate approach to the method of O'Carroll *et al* (2008) is to derive a theoretical uncertainty estimate for the satellite SST (Bulgin *et al*, 2016), which can then be validated using satellite/drifter differences (Lean and Saunders, 2013; Bulgin *et al*, 2016; Neilsen-Englyst *et al*, 2018). The concept of uncertainty validation is presented in detail by Corlett *et al* (2014). Briefly, the standard deviation of the satellite/drifter differences is comprised of contributions from the satellite and drifter measurements, as well
as terms to represent the spatial and temporal differences between the two measurements. Having used models to adjust the drifter measurement to be the same time and depth as the satellite SST, Corlett *et al* (2014) showed the standard deviation of the satellite/drifter differences approximately reduces to two terms, the satellite SST uncertainty and the drifter SST uncertainty as in Equation 1.

$$\sigma_{Satellite\ minus\ drifter} \cong \sqrt{\sigma_{Satellite}^2 + \sigma_{drifter}^2} \qquad (1)$$

Figure 4 shows a comparison between 1 October 2016 and 30 June 2017 of satellite SST validation results for the dual-view 3-channel retrieval from SLSTR for two sets of drifters: all drifters in Fig. 4a, and a subset of HRSST-1 and HRSST-2 drifters in Fig. 4b. In the figure, the green lines indicate the theoretical dispersion of uncertainties using Eq. (1) and a value of 0.20 K for $\sigma_{drifter}$ (an assumption between those of O'Carroll *et al*, 2008 and Lean and Saunders, 2013). The blue lines indicate the calculated dispersion for each set of data and the red lines indicate the standard error. If the assumptions are
correct then the dispersion of the blue lines should track the spread of the green lines, which we see is the case in Fig. 4a (all drifters). Where the dispersion does not match the expected spread, the large standard errors imply a low number of satellite/drifter differences in those bins. For the subset of HRSST drifters, Fig. 4b shows that the dispersion underestimates the spread, even for low standard error cases, meaning one assumption is incorrect in this case.

With all other factors being equal, the distinction in the drifter type between Fig. 4a and Fig. 4b suggests the drifter uncertainty assumed (0.20 K) is inappropriate for the HRSST subset. To verify this, Figure 5 contains the same data as Figure 4 but with the theoretical dispersion (green lines) calculated for a drifter uncertainty of 0.05 K. While the calculated



dispersion does not track any more the expected spread for all drifters (Fig. 5a), the assumption of 0.05 K for the uncertainty of the HRSST drifter data gives a much better fit (Fig. 5b). This demonstrates the improved quality of HRSST drifter data for satellite SST validation.

## 2.5 Limited traceability

Adopting a more general point-of-view for SST observations, several works have already attempted to document the uncertainties in the various *in-situ* SST measurement methods. The present paper does not attempt to review all these efforts, but cites relevant results from the comprehensive review of Kennedy (2014). While the focus of this earlier work was on the creation on long time-series, with the largest issues identified at the time of World War II (transition on ships from bucket to engine-room intake), the quality of SST buoys was found to be the subject of several concerns. The first concern is the

spread in quality between buoys, depending on the source of the uncertainty estimate, with no reliable link to the actual metrological reference. The second concern is a suggested improvement in quality over time, though without quantified evidence or clear *a priori* reason for it that would be explained by metrological documentation. Both points stem from an insufficient knowledge of the sensor technology, and of the calibration procedure that was actually used, for each drifting buoy deployed. The results shown earlier, showing differences in SST quality between general drifters versus HRSST

drifters, reinforce the importance of enhancing the knowledge of drifter metrology and metadata.

## 3. Design of the SVP-BRST

The HRSST-2 efforts were initiated by the cal/val needs of AATSR SST retrievals. With the demise of this instrument after ten years of service in 2012 (ESA Communications Department, 2012), the HRSST-2 developments were put to a halt, until the replacement sensor (SLSTR on Sentinel-3) was launched. However, this gap gave time to finish all HRSST-2

deployments and review the lessons learnt from them. Coupled with the need to assert long consistent time-series of SST at an accuracy level compatible with SLSTR requirements, sound bases were used to imagine a novel platform for reference SST. The result is the SVP-BRST, based on the SVP-B design (Sybrandy *et al*, 2009), but adding the HRSST-2 requirements presented earlier, as well as others, described hereafter.

The first additional requirement is to employ an additional HRSST sensor, in addition to the regular SST sensor. The HRSST sensor collects data within the 5 minutes before the round hour, when the position is updated by means of GNSS. The mean SST is to be computed from 1 Hz SST measurements. In addition, the data can be relayed at 1 Hz frequency for investigation. Furthermore, the distribution of SST observed within the 5 minutes is transmitted at coarse resolution (10% percentile, 30% percentile, 50% percentile or median, 70% percentile, and 90% percentile). This non-parametric information

makes no assumption about the shape of the SST distribution: it can be used to drive an ensemble of applications, rather than using solely the mean SST, and to assess for example whether the SST distribution is symmetric.

Second, the HRSST sensor is removable from the buoy with simple tools (see Fig. 6), and includes a co-located pressure sensor that allows reporting static pressure with an accuracy of 5 cm in calm waters. Even if the instrument is affected by accelerations in wavy conditions, and the depth is only valid in rather calm conditions (when the sensor depth is already

known by design), information can be derived about the hydrostatic water pressure variability (within 5 minutes).

Third, all SST sensors are insulated, to shield them from unwanted effects caused by the non-water surrounding environment. This aims to avoid, for the SST sensors, exchanges by conduction with the buoy hull, exchanges by radiation with the sun and the atmosphere, and radio interference from the buoy electronic board and antenna. This is done in practice

by using, respectively, insulating material between the sensor and the buoy, a small cap to shield the SST sensor from radiation, and a metal plate underneath the buoy electronic board and antenna.

Fourth, the HRSST sensor is defined with a calibrating housing and protocol. Calibration coefficients are determined for each HRSST sensor individually so that their expanded calibration uncertainty can be assessed. These uncertainties are

calculated according to the Guide For Uncertainty of Measurement (BIPM, 2008). They are found to be smaller than 0.01 K for each buoy. Response time and systematic errors related to the integration in the buoy have been assessed on two prototypes. The details of these laboratory measurements will be the subject of another paper.

## 4. Results

Initial testing was conducted in the Brest area (Fig. 7). The results presented hereafter are based on data collected by the two

prototypes between 27 April and 11 June. The data are available in open access (see the section on data availability).

### 4.1 Deployment

Two SVP-BRST prototypes were deployed, as shown in Table 4. At the time of writing, the second prototype is still operating. Before deployment for release, the buoys were deployed briefly on 23 April for comparison in the seawater with an SBE-35 thermometer. The SST differences were then found to be -0.006 K for one buoy and -0.001 K for the other buoy,

thereby meeting the 0.01 K claimed uncertainty. In comparison, the SST difference between the regular (or analogue) SST sensor with the SBE-35 was found to be -0.05 K (for both buoys).

### 4.2 Analysis of the data collected at sea

Once deployed on 26 April 2018, the buoys have followed the tracks shown in Fig. 8. The separation distance between the two buoys, initially under 1 km, remained under 10 km until 23 May. After that, the two buoys quickly diverged until the

first one ran ashore.

The buoy reports data to shore using Iridium according to a binary data format number #091 documented by Blouch *et al* (2018). Besides the usual parameters reported by SVP-B buoys (position, time, strain gauge, air pressure, analogue SST, and other technical parameters such as battery voltage and GNSS Time To First Fix), one notes the following key additions: the mean temperature over 5 minutes reported by the HRSST sensor, 5 percentiles of the SST distribution within that time interval (10%, 30%, 50% or median, 70%, and 90%), and the mean and the standard deviation of the hydrostatic water pressure during 5 minutes.

These parameters are shown in Fig. 9, where atmospheric pressure, SST, and significant wave height from the ECMWF operational analyses have been added. This information was co-located to the buoy dates, times and locations using the same procedure as described in section 2.2 (albeit at different horizontal and temporal resolutions). For the sake of comparing results, the time-series are only for as long as both buoys were freely drifting (until 11 June).

The information from ECMWF analyses, although at a horizontal resolution of around 10 km, is independent from the buoys. It hence provides interesting information to consider when assessing the buoy data. For air pressure (Fig. 9a), both buoys agree with the ECMWF analyses to within 0.8 hPa RMS. This is comparable to state-of-the-art SVP-B deployed in this region.

For SST (Fig. 9b), the comparison to ECMWF analyses only suggests that the latter are typically lagging behind the buoy evolution by 24 hours, until 5 June 2018. It must be remembered that the SST is not currently analyzed in the ECMWF prediction system, but this system was upgraded on June 6, including a component to include atmosphere-ocean coupling (Buizza *et al*, 2018).

The depth inferred from the HRSST hydrostatic pressure sensor (Fig. 9c) shows values around 15 to 18 cm (which is the design location of the HRSST sensor). The spread between the two estimates is stable in time, around 4 cm. The calibration procedure of the pressure sensors may explain this difference. This remains however close to the design depth of 18 cm below the flotation line of the buoy.

The spread in the SST percentiles, shown in Fig. 9d, is usually within 0.1 K but sometimes exceed 0.3 K. In such situations, the calibration accuracy of the sensor is not of much help to help exploit the data for precise comparison with other sources. However, the availability of five estimates of SST, instead of just the mean, should help users move their applications to a small (5-member) ensemble, and better understand how the spread in input *in-situ* SST impacts their products.





Figure 9e shows the standard deviation of depth (inferred assuming hydrostatic equilibrium). This estimate varies between 1.5 and 3.5 cm. It is largest when the significant wave height (estimated by the ECMWF analyses) is largest. This is expected from the buoy dynamics (as the pressure measured will be affected by positive and negative accelerations), and confirms that the ECMWF wave height analysis appears to be correct. Given this result, the larger spread in SST percentiles appears to be well-correlated with situations where the wave heights are smaller. This would seem to validate the conjectures formed earlier by revisiting the HRSST-2 SVP-BS data record, namely that the sea-state is an important parameter to consider when exploiting the *in-situ* SST data.

Regarding the SST data, we see that both buoys capture fairly well the diurnal warming/cooling cycle, a feature that is generally clearly missing from the ECMWF analyses. What is more, the amplitude of the daily cycle is variable, suggesting that the local ocean and atmospheric dynamics impacts the SST measured by the buoys. This is indeed the case for the period from 29 April to 5 May (time period A in Fig. 10): the observed SST is slightly cooler and, crucially, is missing the diurnal cycle found in the rest of the time-series. Looking at co-located wind data (not shown), we do not find any clear modification, suggesting that the reason for this behavior in the SST data is principally oceanic and not atmospheric. Indeed, if we look at the buoys' location during that time period, we see that they are trapped within an eddy core (Fig. 11), and, significantly, it is a cold eddy. It is known that these eddies generate an upwelling within their core, leading to colder and vertically more homogeneous surface and near surface waters. The buoy data suggest that this upwelling more than compensates the diurnal warming and eliminates the near surface stratification. During time period A, the average diurnal cycle measured by the two buoys is rather weak (Fig. 13a and 13b).

Once the buoys move out of the eddy core (Fig. 12), the diurnal cycle is once again found in the data. This is visible in Fig. 10 during time period B, and in Fig. 13c and 13d, where the daily amplitude in SST exceeds 0.5 K (when it was less than 0.2 K in time period A). What is more, during this period of fairly stable SST, we see that the sensor depth is at its greatest (Fig. 9c), suggesting reduced wave activity.

Looking at the evolution of SST 5-minutes percentiles enables to gauge the small-scale variations in temperature near the surface. Figure 14 shows that the two buoys during time period A, as well as the first buoy during time period B, present smaller departures from the mean throughout the day than the second buoy during time period B. The maps in Figures 11 and 12 may hold the clue to explaining this: in the first three cases, the buoys are the closest to eddies, while the fourth situation is when the buoy is travelling furthest from an eddy core. Overall these remarks suggest that the ocean surface circulation may be of importance too, in addition to sea-state, to properly exploit the *in-situ* SST data for satellite cal/val, as this may affect the representativeness of the SST observed *in-situ*.





## 5. Conclusions

Revisiting the previous HRSST drifter initiatives, it was found that higher-quality SST was likely to be collected by such drifting buoys, as compared to general drifters. The following points were also identified to require further consideration, to improve upon HRSST-2 drifters. First, the sea-state dynamics, affected by the wave activity, has influence on the vertical

stratification, so that the depth of the sensors is an important parameter to monitor. Second, the housing of the HRSST sensors needs to be insulated from external influences other than exchanges of heat with the seawater, in order to yield data that reflect the diurnal cycle without the effect of heat conduction from the buoy and heating of the sensor by direct solar radiation. Third, a better-documented protocol is needed for initial sensor calibration, allowing post-mission recalibration, to avoid introducing additional uncertainty through the use of unspecified calibration procedures. Fourth, traceability to

national metrological standards needs to be established.

These findings were taken onboard to design a novel buoy, for the sake of providing FRM SST data for the calibration and validation of satellite SST. The new buoy, called SVP-BRST, carries two SST sensors, one of standard manufacture, the other of absolute uncertainty better than 0.01 K (absolute uncertainty refers here to expanded uncertainty). In addition to

measuring SST with improved calibration, the HRSST sensor also includes a hydrostatic water pressure sensor. The present paper indicates the initial design, which may evolve slightly as experience is gained from expected future deployments in greater numbers.

The two prototypes deployed in the Mediterranean Sea feature, before release, deviations within 0.01 K from a reference

SBE-35 thermometer. Once freely drifting, the buoys observe that the SST spread within 5 minutes is usually smaller than 0.1 K, especially when the sea-state is well-mixed and the buoys are within an eddy core. The availability of percentiles from the 5-minute distribution of SST sampled at 1 Hz (by a sensor with a fast response time) should help users improve their data processing chain to move towards an ensemble approach. The results in this paper suggest that it is important to consider the sea-state mixing and the ocean surface circulation to understand the representativeness of the *in-situ* SST data, as they both

affect observed SST variations (within the day and within 5 minutes). Consequently, they may both be worth considering in the process of satellite SST cal/val.

In addition, a fairly standard analysis, where ocean dynamics behaviour can be inferred from the buoy data, suggests that the high resolution SST data holds a wealth of information. Properly analyzed and interpreted, this data can provide a useful

insight of the dynamics of the sampled area, especially when supplementary information is brought into the picture to consider sea-state and ocean surface circulation. Even more interesting may be to collect full samples of 1 Hz data, when possible, in addition to the summaries of the distribution with 5 percentiles. Such a High-Frequency HRSST dataset



(HFHRSST) may serve other applications beyond satellite SST ca/val, such as fine-scale model developments and enhanced understanding of SST variability.

Future efforts include evaluation of the HRSST sensor drift. This will be done by keeping one SVP-BRST buoy at post in a monitored environment, and by recovering as many SVP-BRST buoys as possible. The goal will be to assess whether the temporal stability of SST from drifting buoys is within +/- 0.01 K/year after manufacture. This is important for climate monitoring, as initial results from past HRSST-2 buoys, presented in this paper, suggest temporal drifts that are systematically negative and close to this figure, though the very small number of drifting buoys surveyed (3) is not significant enough to be conclusive. At least 100 SVP-BRST buoys are expected to be deployed in the next three years, with a view to cover a wide range of atmospheric and oceanographic conditions.

**Code availability**

N/A

**Data availability**

The novel prototype drifter data used in this publication are available in open access: http://doi.org/10.5281/zenodo.1410401

**Sample availability**

N/A

**Appendices**

None

**Supplement link (will be included by Copernicus)**

None

**Author contribution**

Paul Poli drafted the main text of this paper, prepared several of the figures and corresponding scientific analysis, and prepared the manuscript for submission. Marc Lucas drafted the introduction and, along with Gary Corlett, contributed





results and scientific analysis. Arnaud David and Marc Le Menn contributed to the buoy design and calibration. Pierre Blouch, David Meldrum, Anne O'Caroll, Kai Herklotz, and Chris Merchant contributed to the buoy design and scientific analysis.

**Competing interests**

**Disclaimer**

**Special issue statement (will be included by Copernicus)**

**Acknowledgements**

The authors are funded by their respective institutions. Additional support, including the development of the SVP-BRST prototypes and the resulting data analyses, is provided by the European Union's Copernicus programme for funding the
10 development of the SVP-BRST drifting buoys under the project 'Towards Fiducial Reference Measurements from HRSST drifting buoys for Copernicus satellite validation' as part of the TRUSTED project led by CLS, with buoy manufacturing by NKE, calibration by SHOM, coordination of deployments by Meteo France, provision of a tethered reference measurement by BSH, and metadata processing and deployment monitoring visualization tools by JCOMMOPS..

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





| WMO identifier | Deployment basin | HRSST sensor model and S/N | Start date | End date |
|---|---|---|---|---|
| 6200683 | North Atlantic | Digital YSI 46000 10021 | 10/07/2012 | 10/12/2012 |
| 6200686 | North Atlantic | Digital YSI 55032 10030 | 14/07/2012 | 17/11/2012 |
| 4400730 | North Atlantic | Digital YSI 55032 10028 | 16/07/2012 | 10/01/2014 |
| 4400769 | North Atlantic | Digital YSI 55032 10023 | 18/07/2012 | 15/11/2012 |
| 4400775 | North Atlantic | Digital YSI 55032 10022 | 19/07/2012 | 25/01/2014 |
| 4400776 | North Atlantic | Digital YSI 55032 10029 | 20/07/2012 | 18/02/2013 |
| 1300659 | North Atlantic | Digital YSI 55032 10010 | 14/10/2012 | 17/03/2016 |
| 1500545 | South Atlantic | Digital YSI 55032 10006 | 15/10/2012 | 30/08/2013 |
| 1300660 | North Atlantic | Digital YSI 55032 10003 | 22/10/2012 | 02/02/2016 |
| 1300661 | North Atlantic | Digital YSI 55032 10009 | 31/10/2012 | 22/06/2014 |
| 4100738 | North Atlantic | Digital YSI 55032 10001 | 09/11/2012 | 21/02/2014 |
| 4100739 | North Atlantic | Digital YSI 55032 10007 | 13/11/2012 | 21/12/2015 |
| 1500546 | South Atlantic | Digital YSI 46000 10002 | 23/12/2012 | 29/12/2012 |
| 1500547 | South Atlantic | Digital YSI 46000 10005 | 24/12/2012 | 28/06/2015 |
| 1500548 | South Atlantic | Digital YSI 46000 10004 | 26/12/2012 | 28/05/2015 |
| 6200515 | North Atlantic | Digital YSI 55032 10064 | 21/02/2013 | 18/01/2014 |
| 4400770 | North Atlantic | Digital YSI 55032 10050 | 22/02/2013 | 17/04/2014 |
| 6200514 | North Atlantic | Digital YSI 55032 10065 | 22/02/2013 | 20/08/2015 |
| 4400771 | North Atlantic | Digital YSI 55032 10035 | 22/02/2013 | 27/10/2014 |
| 4400550 | North Atlantic | Digital YSI 55032 10027 | 20/03/2013 | 30/01/2014 |
| 1300662 | North Atlantic | Digital YSI 55032 10070 | 12/04/2013 | 15/12/2015 |
| 1300664 | North Atlantic | Digital YSI 55032 10033 | 13/04/2013 | 24/04/2015 |
| 6200712 | North Atlantic | Digital YSI 55032 10063 | 07/05/2013 | 12/01/2014 |
| 6200695 | North Atlantic | Digital YSI 55032 10038 | 07/05/2013 | 03/02/2016 |
| 4400868 | North Atlantic | Digital YSI 55032 10037 | 08/05/2013 | 29/08/2016 |
| 4400604 | North Atlantic | Digital YSI 55032 10047 | 09/05/2013 | 01/07/2013 |
| 1300665 | Tropical Atlantic | Digital YSI 55032 10040 | 27/05/2013 | 04/03/2014 |
| 1300666 | Tropical Atlantic | Digital YSI 55032 10061 | 27/05/2013 | 17/02/2014 |
| 3100718 | North Atlantic | Digital YSI 55032 10068 | 11/06/2013 | 12/11/2016 |
| 3100734 | North Atlantic | Digital YSI 55032 10066 | 05/11/2013 | 15/12/2016 |
| 3100866 | North Atlantic | Digital YSI 55032 0010 | 06/11/2013 | 13/04/2015 |
| 3100868 | South Atlantic | Digital YSI 55032 0008 | 10/12/2013 | 30/01/2017 |



| | | | | |
|---|---|---|---|---|
| 6200537 | North Atlantic | Digital YSI 55032 10032 | 09/06/2014 | 07/03/2015 |
| 4400866 | North Atlantic | Digital YSI 55032 10052 | 21/06/2014 | 03/01/2017 |
| 6500598 | North Atlantic | Digital YSI 55032 10069 | 25/06/2014 | 18/05/2015 |
| 4400871 | North Atlantic | Digital YSI 55032 10034 | 27/06/2014 | 28/01/2016 |
| 1300667 | Tropical Atlantic | Digital YSI 55032 0009 | 03/07/2014 | 26/10/2014 |
| 1300668 | Tropical Atlantic | Digital YSI 55032 0007 | 04/07/2014 | 11/02/2015 |
| 1500549 | Tropical Atlantic | Digital YSI 55032 0006 | 05/07/2014 | 29/03/2015 |
| 4400548 | North Atlantic | Digital YSI 55032 10048 | 11/08/2014 | 16/03/2016 |
| 4400603 | North Atlantic | Digital YSI 55032 10031 | 07/10/2014 | 07/03/2015 |
| 4400604 | North Atlantic | Digital YSI 55032 10055 | 08/10/2014 | 10/02/2017 |
| 4400608 | North Atlantic | Digital YSI 55032 10051 | 10/10/2014 | 08/03/2016 |
| 6200552 | North Atlantic | Digital YSI 55032 10067 | 10/10/2014 | 06/01/2016 |
| 6400551 | North Atlantic (*) | Digital YSI 55032 10064 | 23/06/2015 | 05/02/2018 |
| 4400770 | North Atlantic (*) | Digital YSI 55032 10028 | 02/07/2015 | 30/11/2015 |
| 1501601 | Tropical Atlantic | Digital YSI 55032 10039 | 16/11/2016 | 06/09/2017 |
| 4101711 | North Atlantic | Digital YSI 55032 10036 | 06/07/2017 | 20/10/2017 |

**Table 1**. Mission report of HRSST-2 SVP-B buoys. A star indicates redeployment (note the WMO identifier may have changed, possibly re-using a number previously assigned to an earlier buoy). The third column shows SST sensor references.





| WMO identifier | Deployment basin | HRSST sensor model and S/N | Start date | End date |
|---|---|---|---|---|
| 4100736 | North Atlantic | Digital YSI 46000 10014 | 14/02/2012 | 26/01/2013 |
| 6200513 | North Atlantic | Digital YSI 46000 10011 | 18/03/2012 | 17/01/2013 |
| 6200505 | North Atlantic | Digital YSI 46000 10017 | 25/03/2012 | 10/04/2013 |
| 6200501 | North Atlantic | Digital YSI 46000 10019 | 29/06/2012 | 10/12/2012 |
| 6100788 | Mediterranean Sea | Digital YSI 46000 10020 | 04/09/2012 | 16/02/2013 |
| 3100739 | North Atlantic | Digital YSI 46000 10016 | 30/11/2012 | 06/07/2013 |
| 3100740 | North Atlantic | Digital YSI 46000 10044 | 01/12/2012 | 06/03/2013 |
| 6100530 | Mediterranean Sea | Digital YSI 46000 10013 | 30/01/2013 | 19/05/2013 |
| 6100525 | North Atlantic | Digital YSI 46000 10042 | 22/02/2013 | 16/08/2013 |
| 6100524 | North Atlantic | Digital YSI 46000 10049 | 22/02/2013 | 05/05/2013 |
| 6200504 | North Atlantic | Digital YSI 46000 10045 | 24/05/2013 | 27/11/2014 |
| 1300899 | Tropical Atlantic | Digital YSI 46000 10043 | 26/05/2013 | 10/12/2013 |
| 6200509 | North Atlantic | Digital YSI 46000 10062 | 27/05/2013 | 15/10/2013 |
| 2300587 | Indian Ocean | Digital YSI 46000 10071 | 09/06/2013 | 07/09/2013 |
| 2300588 | Indian Ocean | Digital YSI 46000 10053 | 09/06/2013 | 07/09/2013 |
| 4100737 | North Atlantic | Digital YSI 46000 10059 | 06/12/2013 | 10/03/2015 |
| 4100800 | North Atlantic | Digital YSI 46000 10058 | 06/12/2013 | 16/01/2015 |
| 6200500 | North Atlantic | Digital YSI 46000 10054 | 12/06/2014 | 18/02/2016 |
| 6500511 | North Atlantic | Digital YSI 46000 10056 | 17/06/2014 | 25/06/2014 |
| 3100719 | Tropical Atlantic (*) | Digital YSI 46000 10020 | 11/04/2015 | 20/06/2015 |

**Table 2**. Similar to Table 1, but for HRSST-2 SVP-BS buoys (each buoy was also fitted with a CT probe).





| WMO id. | Lab. # | Date | Mean error | Time interval since lab#1 (days) | Temporal drift since lab#1 |
|---|---|---|---|---|---|
| 4400871 | 1 | 02/10/2012 | -0.010 K | 0 | - |
|  | 2 | 23/09/2016 | -0.063 K | 1452 | -0.013 K/year |
|  | 3 | 16/08/2017 | -0.043 K | 1779 | -0.007 K/year |
| 4400608 | 1 | 16/10/2012 | -0.006 K | 0 | - |
|  | 2 | 23/09/2016 | -0.055 K | 1438 | -0.012 K/year |
|  | 3 | 16/08/2017 | -0.037 K | 1765 | -0.006 K/year |
| 6200552 | 1 | 01/09/2012 | 0.031 K | 0 | - |
|  | 2 | 23/09/2016 | -0.007 K | 1483 | -0.009 K/year |
|  | 3 | 16/08/2017 | +0.014 K | 1810 | -0.003 K/year |

**Table 3**. Individual calibration data for SST sensors from 3 HRSST-2 buoys that were fortuitously recovered. The mean error is the average difference, for several verification points, between the temperature reported by the sensor and the temperature of the calibration bath. Lab. #1 indicates the initial calibration and verification that was made then. The last column, showing temporal drift (in K/year), is 365.25 times the difference between the mean error assessed by lab. #2 (or 3) minus the mean error assessed by lab. #1, divided by the number of days elapsed.



| WMO identifier | Deployment basin | HRSST sensor model and S/N | Start date | End date |
|---|---|---|---|---|
| 6102622 | Mediterranean Sea | Digital MoSens 4658 | 26/04/2018 | 12/06/2018 |
| 6102623 | Mediterranean Sea | Digital MoSens 4656 | 26/04/2018 | - |

**Table 4**. Similar to Table 1, but for 2 prototype SVP-BRST buoys (each buoy is fitted with a HRSST and static pressure probe).





Figure 1: Density plot of the scatter between hull SST measurements (horizontal axis) and CT SST measurements (vertical axis), from HRSST-2 SVP-BS buoys.



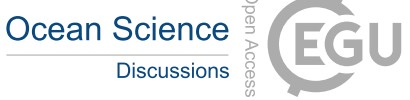

**Figure 2: Differences between the two SST sensors from all HRSST-2 SVP-BS buoys, as a function of (a) solar elevation angle and (b) significant wave height estimated by ERA5.**

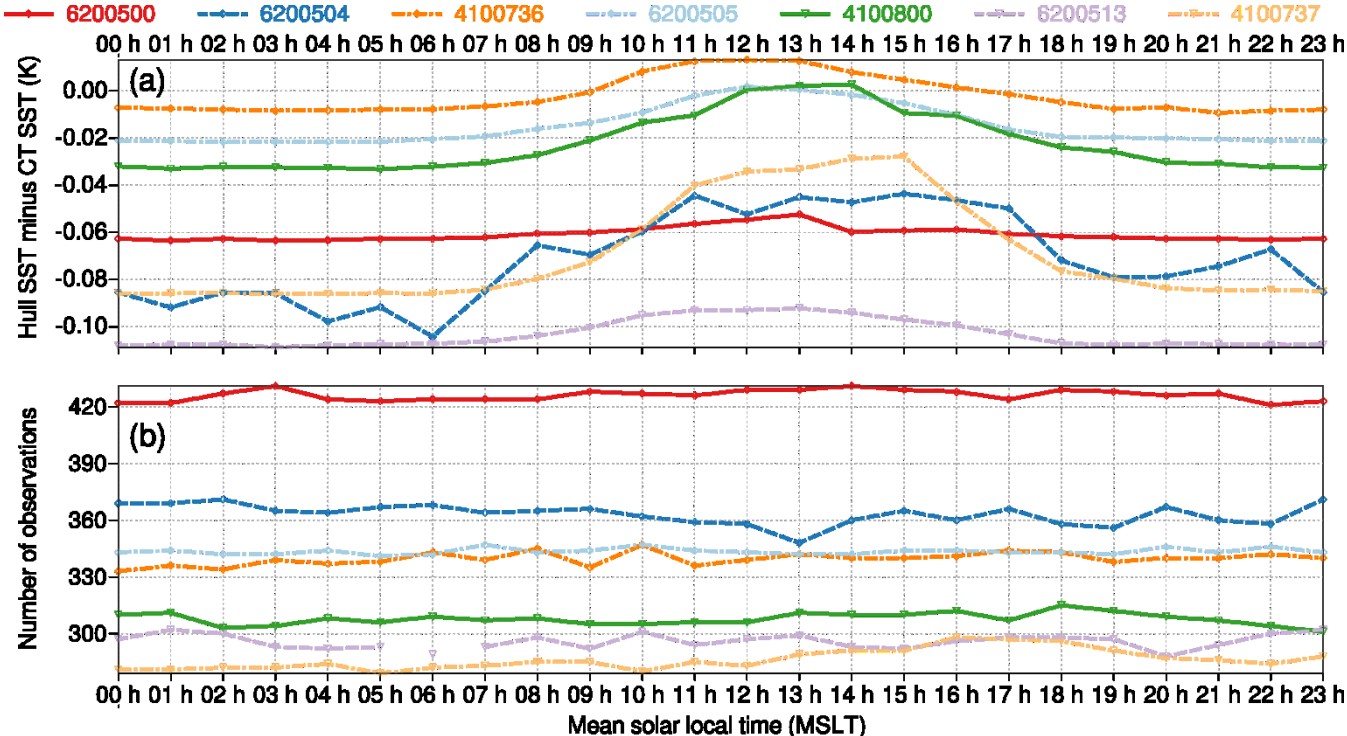

5    **Figure 3: Mean differences (a) between the two SST sensors, with the number of data records shown in (b), for HRSST-2 SVP-BS buoys that reported for at least 250 days (WMO identifier indicated in legend), as a function of mean solar local time (horizontal axis).**





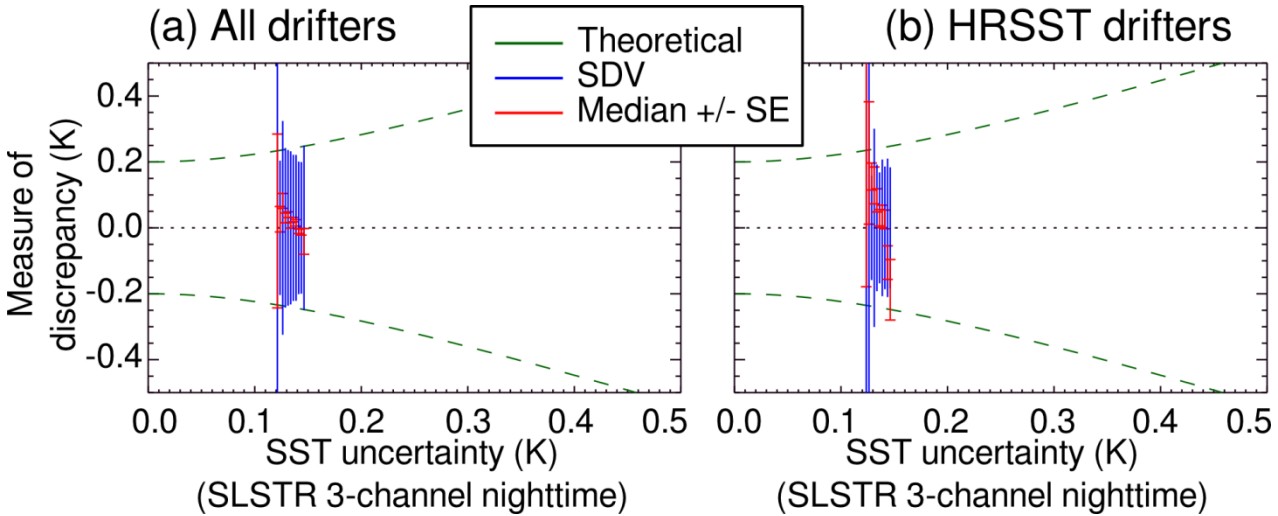

**Figure 4: SLSTR SST uncertainty validation plot for (a) all drifters and (b) a subset of HRSST-1 and HRSST-2 drifters. An uncertainty of 0.20 K is assumed for the drifter SST.**

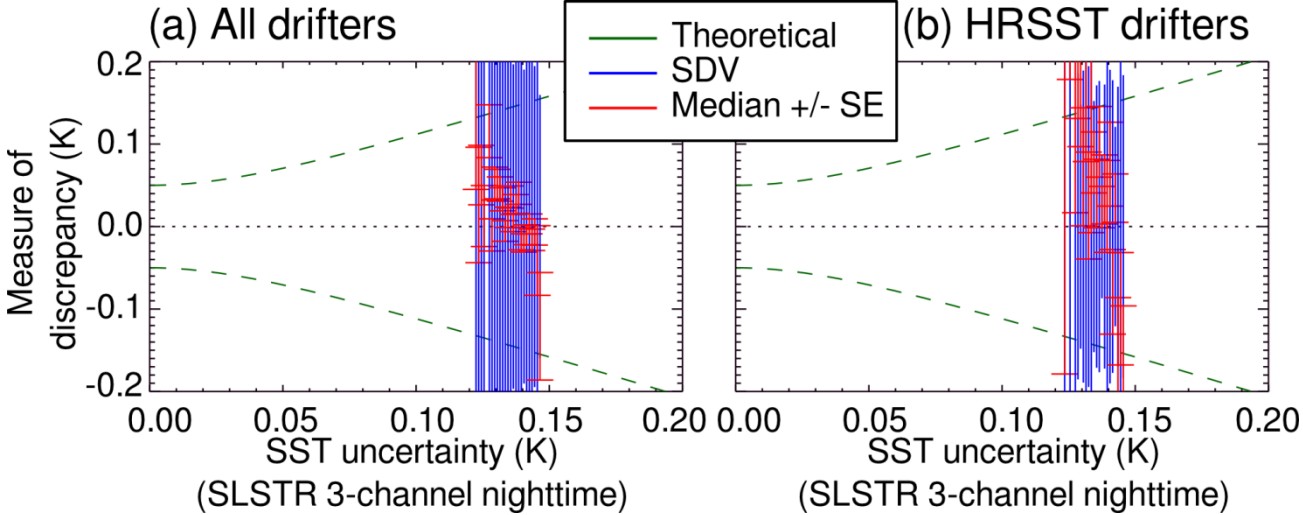

**Figure 5: SLSTR SST uncertainty validation plot for (a) all drifters and (b) a subset of HRSST-1 and HRSST-2 drifters. An uncertainty of 0.05 K is assumed for the drifter SST.**



**Figure 6: Sketch of the SVP-BRST (for the drogue, only the tether attachment is shown here), with the HRSST sensor unplugged shown in zoom (b). Note each SST sensor is protected from solar radiation by a cap.**





**Figure 7: Photo of a SVP-BRST unit deployed for testing in the roadstead of Brest (France).**





**Figure 8: Trajectories of the two SVP-BRST prototypes after deployment on 26 April 2018. The two buoys separated on 22 May 2018. Map data: SIO, NOAA, U.S. Navy, NGA, GEBCO; Map image: Landsat/Copernicus.**





**Figure 9:** **Time-series of data collected by the two SVP-BRST prototypes until one of them ran ashore. Panels (a), (b), and (e) also shows, in lighter colors, ECMWF analyses co-located to the buoys dates, times, and locations.**





**Figure 10: Time-series of the SST data, measured by the two SVP-BRST prototypes' HRSST sensors. A and B indicate two time periods selected for discussion.**





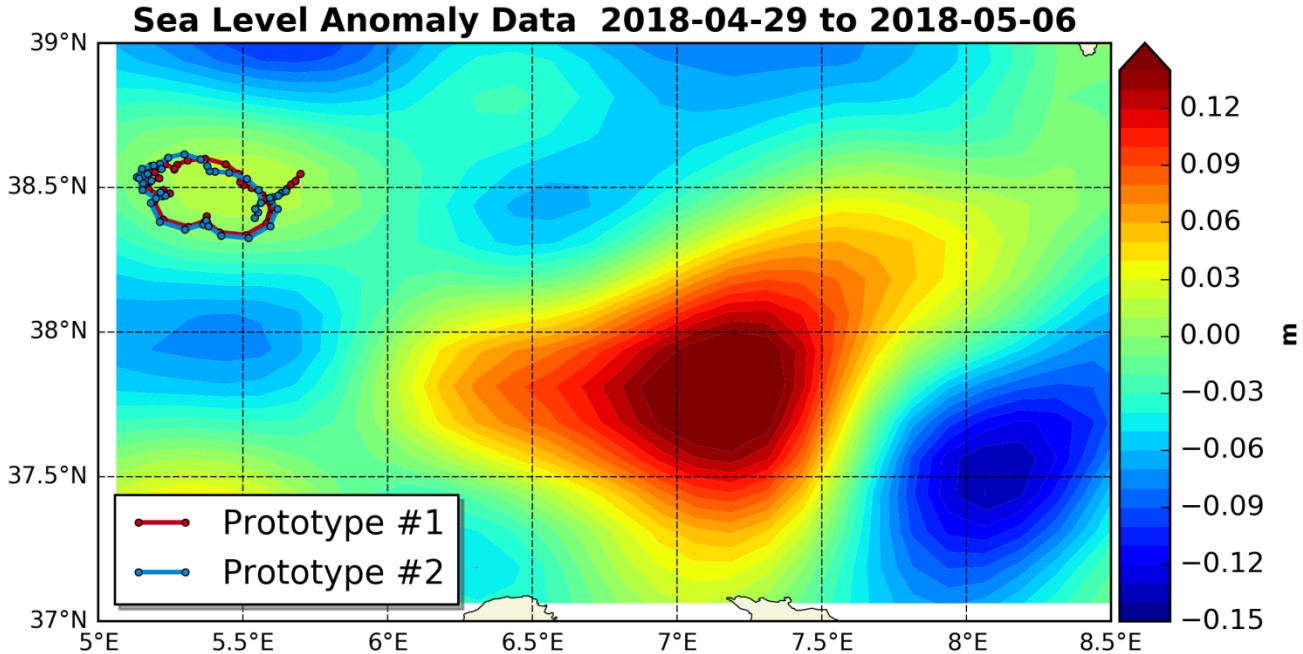

**Figure 11: Mean Sea Level anomaly map with the two SVP-BRST prototypes' tracks overlaid (prototype#1 in red, prototype #2 in blue), for the time period 29 April to 5 May 2018.**



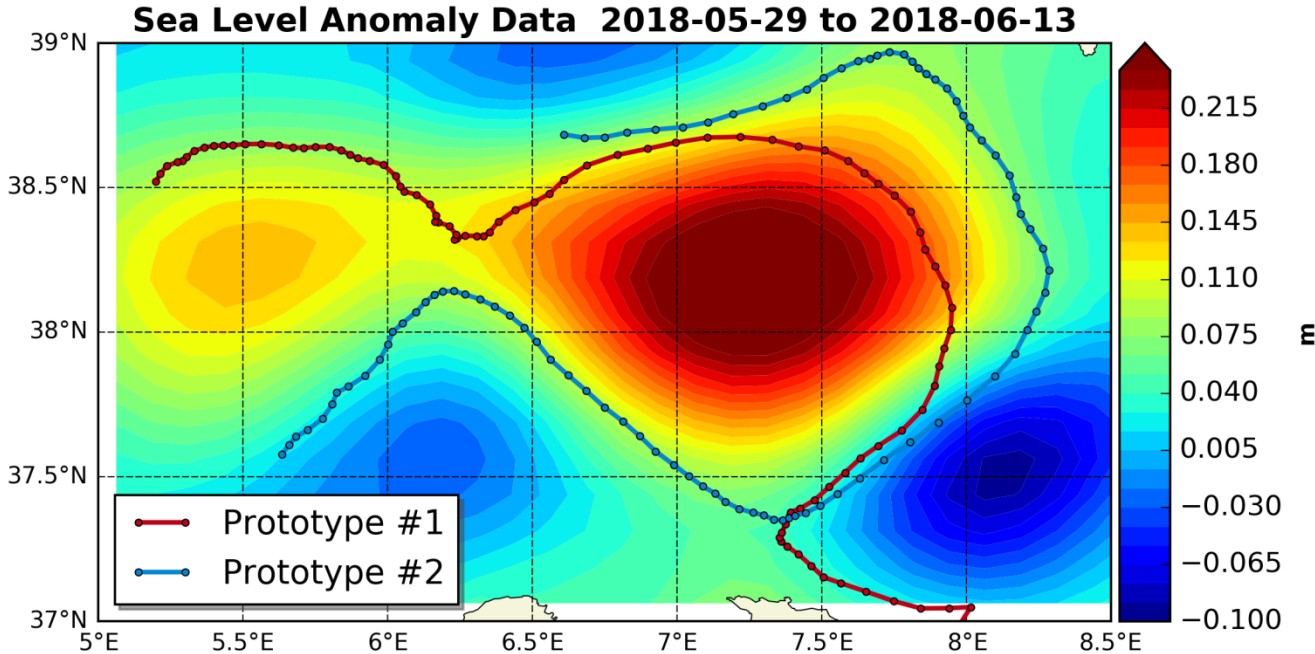

**Figure 12: Mean Sea Level anomaly map with the two SVP-BRST prototypes' tracks overlaid (prototype#1 in red, prototype #2 in blue), for the time period 29 May to 13 June 2018.**



**Figure 13: Average SST diurnal cycle observed by the two SVP-BRST prototypes' HRSST sensors, during time periods A and B defined earlier. For each panel, the reference is the mean SST at 00 UTC. Horizontal thin dotted lines indicate zero.**





**Figure 14:** Diurnal cycle of differences between each 5-minute percentile (five percentiles are reported by the SVP-BRST prototypes: 10%, 30%, 50% or median, 70%, and 90%) and the 5-minute mean. Horizontal thin dotted lines indicate zero and +/- 0.01 K.

