# Peer review of "The Copernicus Surface Velocity Platform drifter with Barometer and Reference Sensor for Temperature (SVP-BRST): Genesis, design, and initial results"

_Ocean Science, 2018_

## Referee Comment (RC1) · Anonymous Referee #1 · 21 Nov 2018

The manuscript is very well written and suffers from few problems needing minor revision. I recommend that it be published after addressing this minor concerns. The only significant addition that the paper needs is a discussion of drogue presence, and how the drogue effects the depth of measurements. This is never brought up in the paper (unless I missed it), but is very relevant and important.

Pg. 3: "Based on lessons learnt, ... a new type of drifter has had to be developed ...". This is overly-harshly worded. It's a new sensor package for an existing type of drifter (the SVP drifter).

[Figure]

Pg. 5: "There are fewer large-magnitude differences are smaller at night ..." Unclear; reword. The authors may wish to note Dong et al. (2017) [http://dx.doi.org/10.1002/2017JC012894] as well, although those differences are for a larger vertical separation of 5m.

Pg. 7: "Recent adjustments have actually recognized buoys as being cooler than ships in terms of SST". Isn't this believed to be due to the sampling bias introduced by ships avoiding intense storms which mix cooler waters to the surface?

Pg. 11: "What is more, during this period of fairly stable SST, we see that the sensor depth is at its greatest". This is confusing. The text had been talking about period B, when SST is far less stable than A. Are they now talking about period A again? The sensor depth appears to be relatively shallow during period B according to Fig. 9c. This sentence needs to be made more clear.
* * *

---

## Author Comment (AC1) · 5 Dec 2018

Thank you for your time in assessing this work and providing constructive comments (reproduced below, in *italics*).

*The manuscript is very well written and suffers from few problems needing minor revision. I recommend that it be published after addressing this minor concern. The only significant addition that the paper needs is a discussion of drogue presence, and how the drogue [a]ffects the depth of measurements. This is never brought up in the paper*

[Figure]

*(unless I missed it), but is very relevant and important.*

The need to discuss drogue presence and how it affects the depth of measurements is duly taken into account. However, the data from the two SVP-BRST prototypes deployed so far are not sufficient to document this effect. Consequently, to look into this, we revisited the data record from previous HRSST-2 buoys, most particularly the SVP-BS, since they presented the advantage of providing two high-quality SST measurements located at different depths. As a result, we propose to insert a new section before "2.5 Limited traceability" (text proposed is in **bold**):

**2.5 Influence of the drogue on drifter SST measurements**

**This section investigates the effect of the sea anchor or drogue on drifter SST measurements. By exerting its own weight and by following currents centered at 15 meter depth, the drogue pulls the float downwards, via the tether. This maintains the float and its drogue aligned in the vertical, in wave troughs. When the drogue is lost, the float has more freedom to oscillate by roll and pitch, and the temperature probe can sometimes be exposed to waters closer to the surface. Also, when in that situation, the float is more likely to reach wave crests. There, the sky visibility is improved, reducing the GPS Time To First Fix (TTFF), which can serve as an additional indicator of drogue loss (Petolas, 2013).**

**To investigate the influence of the drogue, the SVP-BS data record is revisited. These buoys used submergence sensors, whereas drifters nowadays use strain gauges, e.g. as indicated by Rio (2012), who developed an advanced method to identify drogue loss using drifter currents, satellite altimetry, and wind re-analysis data. The submergence (or tether strain gauge) readings are neither straightforward to interpret, nor fully reliable on their own (Rio, 2012). However, the SVP-BS drifter data considered here (available from the Coriolis In-Situ Thematic Assembly Centre) are not found in the drifter dataset of Rio and Etienne (2018), which includes drogue presence flags. Consequently, for this analysis,**

we use the submergence and GPS TTFF data. A visual inspection indicates that 10 of the 20 buoys in Table 2 have lost their drogues during their mission. For these buoys, two series of data records are extracted: (1) before drogue loss, and (2) after drogue loss.

During day time, the median of the differences between the twin SST measurements is -0.04 K in (1), whereas it is -0.03 K in (2). The reduction in differences may appear insignificant, but it is consistent with the CT sensor being more often exposed to depths similar to the sensor integral to the hull when the drogue is lost, than when the drogue is present. Similarly, the robust standard deviation of the differences between the twin SST measurements is 0.03 K in (1), whereas it is 0.01 K in (2). Again, this reduction is consistent with drogue loss, for the same reasons.

During night time, no influence of the drogue loss is expected, if the temperatures are homogeneous just below the surface. This is indeed what is observed. The median of the differences is -0.04 K in both (1) and (2), and the robust standard deviation of the differences is 0.03 K in both (1) and (2).

In other terms, the SVP-BS data record confirms the expectation that once the drogue is lost, the SST probes on a drifter are more likely to be exposed to water immediately below the surface, than when the drogue is present, and this effect is more visible in the presence of stratification (e.g., during day-time). To keep track of the drogue effect on SST measurements, it is important to monitor drogue loss as well the immersion depth and its variations.

In addition, we propose to replace the following sentence in section 3:

"The result is the SVP-BRST, based on the SVP-B design (Sybrandy et al, 2009), but adding the HRSST-2 requirements presented earlier, as well as others, described hereafter."

by

**"The result is the SVP-BRST, based on the SVP-B design (Sybrandy et al, 2009), with a strain gauge to detect drogue loss. In addition, the HRSST-2 requirements presented earlier are included, as well as others, described hereafter."**

*Pg. 3: "Based on lessons learnt, ... a new type of drifter has had to be developed ...". This is overly-harshly worded. It's a new sensor package for an existing type of drifter (the SVP drifter).*

Indeed, this sentence will be modified as suggested (in other places, too). However, the data to be reported by SVP-BRST will eventually feature also 5-minute samples of 1-Hz data.

*Pg. 5: "There are fewer large-magnitude differences are smaller at night..." Unclear; reword. The authors may wish to note Dong et al. (2017) [http://dx.doi.org/10.1002/2017JC012894] as well, although those differences are for a larger vertical separation of 5m.*

We propose to reword this sentence as follows:

**The differences are smaller at night and when the Sun is more than 30 degrees below the horizon.**

Also, a reference to Dong et al. (2017) will be added as suggested. We note that the differences found by Dong et al. are indeed for a different layer, and not the top-most levels where the effects of cooling by wind (in normal conditions) may be expected, and where one expects a reversal in the water temperature profile in the top decimeters in the case of day time, light winds (consistent with the GHRSST schematic defining SST foundation). However, a reference to Reverdin et al. (2013), who had investigated the issue of temperature stratification effects seen by drifters, will be added.

*Pg. 7: "Recent adjustments have actually recognized buoys as being cooler than ships in terms of SST". Isn't this believed to be due to the sampling bias introduced by ships*

*avoiding intense storms which mix cooler waters to the surface?*

We were not referring here to the sampling bias, but to the rather well-documented result that ship SSTs tend to be warmer than drifter SSTs, even when considering co-located measurements: to support this, we propose to add references to Emery et al. (2001) and Rayner et al. (2010).

*Pg. 11: "What is more, during this period of fairly stable SST, we see that the sensor depth is at its greatest". This is confusing. The text had been talking about period B, when SST is far less stable than A. Are they now talking about period A again? The sensor depth appears to be relatively shallow during period B according to Fig. 9c. This sentence needs to be made more clear.*

There is indeed some confusion in the text at this point. We propose to remove that sentence ("What is more. . ."), as it brings the discussion back to time period A, at a point when the text has moved to discussing time period B.

References to be added to the revised paper:

Dong, S., Volkov, D., Goni, G., Lumpkin, R., and Foltz, G.R.: Near‐sur-face salinity and temperature structure observed with dual‐sensor drifters in the subtropical South Pacific, J. Geophys. Res. Oceans, 122, 5952–5969, doi:10.1002/2017JC012894, 2017.

Emery, W.J., Baldwin, D.J., Schlüssel, P., and Reynolds, R.W.: Accuracy of in situ sea surface temperatures used to calibrate infrared satellite measurements, J. Geophys. Res., 106, 2387-2405, doi:10.1029/2000JC000246, 2001.

Petolas, B.: Status and Performance of MetOcean Iridium Drifting Buoys, DBCP-29 Sci. Tech. Workshop, https://www.jcomm.info/index.php?option=com$_o$etask $=$ $viewDocumentRecord doc ID = 11847[Last accessed 27 November 2018]$, 2013.

Rayner, N.A., Kaplan, A., Kent, E.C., Reynolds, R.W., Brohan, P., Casey, K.S., Kennedy, J.J., Woodruff, S.D., Smith, T.M., Donlon, C., Breivik, L-A., Eastwood, S.,

Ishii, M. and Brandon, T.: Evaluating Climate Variability and Change from Modern and Historical SST Observations. Hall, J., Harrison, D.E. and Stammer, D. (eds.) In Proceedings of OceanObs'09: Sustained Ocean Observations and Information for Society, Vol. 2, 819-829, European Space Agency, 2010.

Reverdin, G., Morisset, S., Bellenger, H., Boutin, J., Martin, N., Blouch, P., Rolland, J., Gaillard, F., Bouruet-Aubertot, P., and Ward, B.: Near–Sea Surface Temperature Stratification from SVP Drifters. J. Atmos. Oceanic Technol., 30, 1867-1883, doi:10.1175/JTECH-D-12-00182.1, 2013.

Rio, M.: Use of Altimeter and Wind Data to Detect the Anomalous Loss of SVP-Type Drifter's Drogue, J. Atmos. Oceanic Technol., 29, 1663-1674, doi:10.1175/JTECH-D-12-00008.1, 2012.

Rio, M.-H., and Etienne, H.: Global Ocean delayed mode in-situ observations of ocean surface currents, SEANOE, doi:10.17882/41334, 2018.

---

## Referee Comment (RC2) · Anonymous Referee #2 · 22 Dec 2018

The submitted paper presents the new generation Copernicus drifting buoy, which has a reference sensor for SST measurements providing more accurate results than standard drifters. The paper is well written and on time for the validation of the SLSTR SST. Thus, it deserves publication in Ocean Science once the following comments are addressed. Although, there is in practice one major comment about the wind speed dependence, which requires a little bit of extra work, this is straightforward and it is not expected to cause difficulties to the authors.

Major comments

1) Page 5, Lines 14-15. Although, the dependence on significant wave height is very interesting and useful, the majority of past SST studies (e.g. Donlon et al. (2002), Morak-Bozzo et al. (2016)) have used the wind speed as the parameter from which the sea-state mixing can be deduced. Thus, it is important to make the link with previous studies and add a panel to Figure 2 of the SST difference against wind speed (e.g. taken from ERA-5) and discuss it in the text.

2) Page 11, Line 7. Similar to the previous comment, add a wind speed panel in Figure 9 and respective discussion in the text.

Minor comments

3) Page 2, Lines 12-15: This sentence is not clear given that AATSR, the precursor of SLSTR, it was also a dual-view radiometer with two on board blackbodies for the calibration of the TIR channels. Please rephrase or clarify.

4) Page 2, Line 20: "... earlier generations of sensors." Please indicate the sensors that are talking about.

5) Page 5, Line 9: "There are fewer ... at night and ...". Please rephrase as it does not make sense.

6) Page 5, Line 12: Please add a comment about the differences at $\pm 1$ K in Figure 2. Also, it is not clear if there are differences larger than $\pm 1$ K or not. Please clarify.

7) Page 6, Line 13: The reference to Table 3 is missing. Probably, it would be better to move the sentence of lines 18-20 a few lines above, as there is reference to Table 3 in line 13, but a description/presentation of Table 3 is introduced below.

8) Page 7, Line 3: It would be useful to add a comment about the above average lifetime of the 3 buoys in Table 3 and also interpret the trends under this light. The drifters have a mean lifetime of $\sim 450$ days, e.g. Lumpkin et al. (2012).

9) Page 7, Lines 9-10: This sentence is true if the assumptions behind the analysis

are true. For example the representativity error is not taken into account. Gruber et al. (2016) provide a mathematical framework indicating how triple collocation penalizes the point instrument (in this case the drifters). This is in line also with the results of the authors e.g. in Figure 9d for differences occurring only within 5 minutes for which the percentiles have been calculated.

10) Figures 4 and 5 (or in the text): Please provide the number of match-ups and the step of the SST uncertainty.

11) Page 8, Line 29: Why not the 25 and 75 percentile (instead of 30 and 70) from which the interquartile range (IQR) can be calculated?

Technical comments

12) Is Figure 7 useful?

13) Page 9, Line 20: Probably add the clarification that the drifters have been in the Mediterranean, as it is confusing with the previous sentence.

14) Page 8, Lines 23-24: Deployment in Brest? Please clarify.

15) Page 11, Lines 19 and 21: Change the numeration of the figures, as Figure 13 appears in the text before Figure 12.

Extra references

Donlon, C.J., P.J. Minnett, C. Gentemann, T.J. Nightingale, I.J. Barton, B. Ward, and M.J. Murray, 2002: Toward Improved Validation of Satellite Sea Surface Skin Temperature Measurements for Climate Research. J. Climate, 15, 353–369, https://doi.org/10.1175/1520-0442(2002)015<0353:TIVOSS>2.0.CO;2

Gruber, A, C-H Su, S Zwieback, W Crow, Wouter Dorigo, and W Wagner. 2016. "Recent Advances in (soil Moisture) Triple Collocation Analysis." International Journal of Applied Earth Observation and Geoinformation 45: 200–211.

[Figure]

Lumpkin, R., N. Maximenko, and M. Pazos, 2012: Evaluating Where and Why Drifters Die. J. Atmos. Oceanic Technol., 29, 300–308, https://doi.org/10.1175/JTECH-D-11-00100.1

---

## Author Comment (AC2) · 17 Jan 2019

We are grateful to Reviewer2 for the time and efforts assessing our work, and for providing comments.

*These comments are reproduced below, in italics.*

**Modifications proposed to the manuscript are shown in bold.**

*The submitted paper presents the new generation Copernicus drifting buoy, which has*

[Figure]

*a reference sensor for SST measurements providing more accurate results than standard drifters. The paper is well written and on time for the validation of the SLSTR SST. Thus, it deserves publication in Ocean Science once the following comments are addressed. Although, there is in practice one major comment about the wind speed dependence, which requires a little bit of extra work, this is straightforward and it is not expected to cause difficulties to the authors.*

Indeed, inclusion of wind speed results could easily be done. (see later)

*1) Page 5, Lines 14-15. Although, the dependence on significant wave height is very interesting and useful, the majority of past SST studies (e.g. Donlon et al. (2002), Morak-Bozzo et al. (2016)) have used the wind speed as the parameter from which the sea-state mixing can be deduced. Thus, it is important to make the link with previous studies and add a panel to Figure 2 of the SST difference against wind speed (e.g. taken from ERA-5) and discuss it in the text.*

Indeed, the relationship between wind and waves is strong and well-established (e.g., Beaufort scale). Figure 2 has been modified to include a panel on wind speed (collocated from ERA5). The following text is proposed for inclusion in section 2.2, just after "information about sea-state":

**"In past SST studies, wind speed is generally used to describe sea-state mixing (e.g., Donlon et al., 2002, Morak-Bozzo et al., 2016). In this study, we also consider significant wave height. Information about both parameters can be obtained by co-locating. . ."**

The in-text discussion of Figure 2 results is also proposed to be modified: **"when the wind speed is up to moderate (under 8-10 m/s) and when the wave heights are up to moderate (under 2-3 m). . ."**

*2) Page 11, Line 7. Similar to the previous comment, add a wind speed panel in Figure 9 and respective discussion in the text.*

A panel was added in Figure 9, showing wind speed collocated from ECMWF operational analyses. The discussion of that figure in section 4.2 is proposed to be modified as follows: "It is largest when the significant wave height (estimated by the ECMWF analyses) is largest, **in line with stronger winds at the same times**." Wind is also to be mentioned in two other occurrences in the same paragraph.

*3) Page 2, Lines 12-15: This sentence is not clear given that AATSR, the precursor of SLSTR, it was also a dual-view radiometer with two on board blackbodies for the calibration of the TIR channels. Please rephrase or clarify.*

We agree with this remark, so we propose to change the text to "**gives it comparable accuracy to similar sensors**".

*4) Page 2, Line 20: "... earlier generations of sensors." Please indicate the sensors that are talking about.*

We propose to remove ", at a higher level than earlier generation of sensors".

*5) Page 5, Line 9: "There are fewer ... at night and ...". Please rephrase as it does not make sense.*

This sentence is proposed to be rephrased as follows: **"The differences are smaller at night and when the Sun is more than 30 degrees below the horizon"**

*6) Page 5, Line 12: Please add a comment about the differences at 1 K in Figure 2. Also, it is not clear if there are differences larger than 1 K or not. Please clarify.*

There are about 87,000 data points in input of Figure 2 (information indicated in the text). To avoid hiding any information, the figure shows at the minimum and maximum all the points that fall outside the range. This represents 154 data points at or below -1 K and 289 data points at or above 1 K. This information is proposed to be added in the text, as follows: "**Differences that are out-of-range (below -1 K or above 1 K) are also shown for completeness (at -1 K and +1 K, respectively); they represent about 0.5% of the entire data record**."

*7) Page 6, Line 13: The reference to Table 3 is missing. Probably, it would be better to move the sentence of lines 18-20 a few lines above, as there is reference to Table 3 in line 13, but a description/presentation of Table 3 is introduced below.*

This change will be made.

*8) Page 7, Line 3: It would be useful to add a comment about the above average lifetime of the 3 buoys in Table 3 and also interpret the trends under this light. The drifters have a mean lifetime of approx. 450 days, e.g. Lumpkin et al. (2012).*

The 3 buoys mentioned in Table 3 had lifetimes that can be found by look-up in Table 1. The following comment is proposed to be added in section 2.3, as follows:

**"The three recovered buoys achieved lifetimes of (respectively) 580, 515, and 453 days (see Table 1). These durations are close to or above the average drifter lifetime of 450 days (Lumpkin et al., 2012). Considering all the estimated temporal drifts shown in Table 3, the temperature biases of these drifters (averaged over the mission duration) would range between -0.002 K and -0.010 K."**

*9) Page 7, Lines 9-10: This sentence is true if the assumptions behind the analysis are true. For example the representativity error is not taken into account. Gruber et al. (2016) provide a mathematical framework indicating how triple collocation penalizes the point instrument (in this case the drifters). This is in line also with the results of the authors e.g. in Figure 9d for differences occurring only within 5 minutes for which the percentiles have been calculated.*

The referee is correct - but the analysis presented later in the paper, based on Corlett et al (2014), does not use triple-collocations, where the point is space (representativeness) contribution is explicitly included. Consequently, on this comment, the text does not need modifying.

*10) Figures 4 and 5 (or in the text): Please provide the number of match-ups and the step of the SST uncertainty.*

"All drifters" relate to 15,551 matchups, and "HRSST" relate to 625 matchups. The uncertainty bins are 0.01 in the upper plot and 0.001 in the lower plot. This information is proposed to be added in the manuscript.

*11) Page 8, Line 29: Why not the 25 and 75 percentile (instead of 30 and 70) from which the interquartile range (IQR) can be calculated?*

There was only very limited space in the Iridium message to include additional information (without incurring increased transmission costs, and hence also increased battery consumption costs, with longer messages). The percentiles were chosen to document as much as possible the potential asymmetry of the measured distribution, at even spacing, hence the 20% increments, starting at the 10th percentile. However, as indicated in the conclusion section, we hope to deploy at a later stage a (limited) number of buoys that will report the full set of 1-Hz measured data (at full resolution of 0.001 K), which will yield even more information for detailed investigations.

*12) Is Figure 7 useful?*

We acknowledge that Figure 7 does not show any scientific results. It has been moved to supplement material. This results in figures being renumbered, from former figure 8 onwards.

*13) Page 9, Line 20: Probably add the clarification that the drifters have been in the Mediterranean, as it is confusing with the previous sentence.*

This information is proposed to be added for clarity, in the first paragraph of section 4 (noting that Table 4 cited just after clearly mentions the deployment area).

*14) Page 8, Lines 23-24: Deployment in Brest? Please clarify.*

The deployment area is now clearer after the modification above.

*15) Page 11, Lines 19 and 21: Change the numeration of the figures, as Figure 13 appears in the text before Figure 12.*

The two figures have now been swapped.

The PDF attachment includes the paper with all changes (in track-changes) as well as the new supplement.

The following references are to be added to the revised paper:

Donlon, C.J., P.J. Minnett, C. Gentemann, T.J. Nightingale, I.J. Barton, B. Ward, and M.J. Murray, 2002: Toward Improved Validation of Satellite Sea Surface Skin Temperature Measurements for Climate Research. J. Climate, 15, 353–369, https://doi.org/10.1175/1520-0442(2002)015<0353:TIVOSS>2.0.CO;2

Lumpkin, R., N. Maximenko, and M. Pazos, 2012: Evaluating Where and Why Drifters Die. J. Atmos. Oceanic Technol., 29, 300–308, https://doi.org/10.1175/JTECH-D-11-00100.1

Please also note the supplement to this comment:
https://www.ocean-sci-discuss.net/os-2018-109/os-2018-109-AC2-supplement.pdf

[Figure]

**Supplement:**

**The Copernicus Surface Velocity Platform drifter with Barometer and Reference Sensor for Temperature (SVP-BRST): Genesis, design, and initial results**

Paul Poli1, Marc Lucas2, Anne O'Carroll3, Marc Le Menn4, Arnaud David5, Gary K. Corlett6, Pierre
5 Blouch7, David Meldrum8, Christopher J. Merchant9, Mathieu Belbeoch10, Kai Herklotz11

[revised manuscript text omitted]

---

## Author Response (AR1)

Author responses to comments 1 and comments 2 have been posted on-line on the article discussion page.

In both cases, all comments have been included as well responses.

A version of the article with all changes highlighted in track-changes mode follows hereafter.

[revised manuscript text omitted]